**Data Availability Statement:** All material for this study are available on OSF: https://osf.io/xrmkn/.

**Funding:** Funding for this research was provided by grants to the first author (A.J.) from the Duke Office of Undergraduate Research Support, as well

# General and specific stress mindsets: Links with college student health and academic performance

**Anna Jenkins**[iD]¤, **Molly S. Weeks**[iD], **Bridgette Martin Hard**[iD]*

Department of Psychology & Neuroscience, Duke University, Durham, NC, United States of America

¤ Current address: Mayo Clinic Alix School of Medicine, Scottsdale, AZ, United States of America
* bridgette.hard@duke.edu

## Abstract

The goal of this cross-sectional, correlational study was to evaluate (a) whether beliefs about stress as enhancing versus debilitating (i.e., *stress mindsets*) vary across sources of stress that differ in duration (acute vs. chronic) and controllability, and (b) how general and source-specific stress mindsets relate to health and academic performance. College students (*n* = 498) self-reported their general and source-specific stress mindsets, perceived distress, health, coping, and GPA. Stress mindsets varied as a function of duration and controllability, and general stress mindsets were only weakly associated with source-specific mindsets. Consistent with previous research, general stress mindsets were associated with health, but some source-specific mindsets were more predictive of health than others—viewing stress from chronic controllable sources as debilitating was most predictive of poor mental and physical health. Measures of stress were also associated with health, and this association was moderated by stress mindsets, suggesting that viewing stress as enhancing can provide a psychological "buffer" against the negative effects of stress. Approach coping and perceived distress were examined as potential mediators of the links between stress mindset and health. Viewing stress as enhancing was related to greater use of approach coping and lower perceived distress, which in turn was related to better health. This research suggests that stress mindset interventions may benefit students' health, and that interventions targeting mindsets for chronic controllable sources of stress may be more effective than general stress mindset interventions.

## Introduction

*There is nothing either good or bad, but thinking makes it so.* (Shakespeare, trans. 1992, 2.2.268–270). A wealth of psychological research supports this statement, underscoring the power of beliefs in influencing behavior, achievement [1, 2], and health outcomes [3]. Recently, research on the power of beliefs has expanded to the domain of stress. People have varying beliefs about stress: some perceive it to be motivational, beneficial, and "good"; whereas others perceive stress to be harmful, taxing, and "bad." Neither belief is unfounded—

as a philanthropic gift to first author (A.J.) from the Charles Lafitte Foundation to the Department of Psychology & Neuroscience at Duke University. There are no grant numbers associated with the funding for this project. The funders had no role in study design, data collection and analysis, decision to publish, or preparation of the manuscript.

**Competing interests:** The authors have declared that no competing interests exist.

stress has been associated with increased focus, productivity, and psychological growth [4], but also with greater susceptibility to physical and mental illness and increased risk of mortality [5–7]. These meta-cognitive outlooks on stress are called "stress mindsets" [4]. These mindsets matter.

Research suggests that stress may affect people differently based on their beliefs about stress. Believing stress to be beneficial has been related to more adaptive cortisol response, increased desire for feedback [4], higher positive affect, and increased cognitive flexibility in the face of stress [8]. Furthermore, beliefs about stress can change: stress mindset interventions that shift valuations from "stress is debilitating" to "stress can be enhancing" have been found to improve coping [9], cortisol reactivity [4], and psychological thriving [8] under stress.

Recent research has investigated the power of stress mindsets in a sample of people who experience especially high levels of stress: college undergraduates [10]. Keech, Hagger, O'Callaghan, and Hamilton [11] found that undergraduates who hold a stress-is-enhancing mindset experience better physical and psychological well-being, as well as stronger academic performance than those who hold a stress-is-debilitating mindset. In the present research, we aimed to strengthen our understanding of stress mindsets in college students as a first step towards developing stress mindset interventions for this population.

In the sections that follow, we first summarize key findings regarding the importance of interpretations and beliefs about stress, including correlational and experimental work demonstrating the influence of stress mindsets on performance and health. We next discuss the relevance of stress-mindset interventions for undergraduates and important gaps in the literature which underscore the goals of the present study, including examining (a) whether stress mindsets vary across sources of stress, (b) whether student stress mindset is related to academic performance and physical and mental health, (c) whether stress mindsets buffer students from the negative impacts of stress on health, and (d) whether approach coping and perceived distress may operate as mediators of the association between stress mindset and student health.

## Is stress good or bad? Stress valuation, appraisal, and mindset

In his landmark book *Stress without Distress*, Hans Selye [12] defined stress as the "nonspecific response of the body to any demand made upon it" (p. 7). A stressor is therefore anything which places demand on the body, whether pleasant or unpleasant. Selye contended that stressors are ever-present, and that stress is something that cannot and should not be avoided —"complete freedom from stress is death" (p. 32). Although Selye argued that stress, a nonspecific response, can be positive (*eustress*) or negative (*distress*), it is more typically equated with distress. The belief that stress is "bad" is pervasive and is supported by a wealth of research showing that stress is related to greater susceptibility to physical and mental illness [13]. Indeed, much research on stress uses measures that equate stress with distress, asking participants how overwhelmed and out-of-control they feel in reference to the demands of their day-to-day lives [14]. Given this dominant view that stress is bad, it is not surprising that many stress-related interventions aim to reduce stress (for a review, see [15]). Yet research also supports Selye's notion that stress can be positive: stress has been related to increased personal initiative and productivity [16] and even growth and personal development [17]. How can stress be both good and bad, and what determines its positive or negative impact? Are there alternatives to stress reduction interventions that can harness stress's potentially positive effects?

Crum, Jamieson, and Akinola [18] proposed redefining stress to disconnect the term from its "bad" reputation and restore Selye's conceptualization that stress can be positive or negative. They define *stress* as the anticipation and experience of encountering a demand, separate from the *stress response*, which describes the body's nonspecific response (physiological,

behavioral, and emotional) to this experience. *Stressors* are then aspects of one's life, such as events or circumstances, that can cause stress. By defining stress in this way, as distinct from its causes (stressors) and outcomes (the stress response), new avenues open for regulating stress and optimizing the stress response. Even if a stressor is perceived as "bad" (e.g., the coronavirus pandemic), the stress resulting from it could be perceived as "good," for example by motivating an individual to help those affected in their community. This positive *valuation* of stress can then shift the stress response in a positive direction, toward what Selye conceived as *eustress* rather than *distress*. This framework opens the possibility of interventions that focus not on reducing stress, but rather on changing stress valuation to optimize stress regulation, such as encouraging people to think of how stress can be utilized, rather than avoided. An additional benefit of altering stress valuations is potentially reducing anxiety about the assumed harm of stress and the need to avoid it, or "stress about stress" [19].

Two types of interventions have been developed to optimize stress by changing stress valuation: stress reappraisal and stress mindset interventions. Stress *reappraisal* interventions involve changing the valuation or *appraisal* of the physiological arousal that occurs as part of the stress response [18]. For example, individuals encouraged to reappraise their arousal related to public speaking as adaptive rather than harmful showed improved performance and less negative affect during a public speaking task, as compared to controls, and the benefit of reappraisal was seen even in participants who began the intervention with low social anxiety [20]. Whereas stress reappraisal interventions focus on valuations of arousal during a specific situation, stress *mindset* interventions target valuations about the general nature and experience of stress [18]. As noted earlier, stress-is-enhancing mindsets (believing stress to be beneficial) have been related to more adaptive cortisol response, increased desire for feedback [4], higher positive affect, and increased cognitive flexibility [8] in the face of stress. Video interventions that shift valuations from "stress is debilitating" to "stress can be enhancing" have been found to improve coping [9] and workplace productivity [8].

## Stress mindsets in the college context

One population that might particularly benefit from stress-optimization interventions are undergraduate students. Undergraduates generally report high levels of stress, and those who report more stress tend to experience poorer physical and mental health [10, 21, 22]. Yet stress is an important and even necessary part of the undergraduate experience. College is filled with challenging intellectual and social experiences, including setbacks, and the process of confronting these stressful experiences can ultimately contribute to student self-discovery and growth [23]. Therefore, putting too much emphasis on eliminating stress from students' lives may be both impractical and counterproductive to the goals of higher education. A focus on stress optimization, rather than reduction, may be a more productive strategy. Prior work supports the benefits of one type of stress optimization, reappraisal of the stress response, as a tool to improve undergraduates' academic performance. Recall that reappraisal interventions differ from stress mindset interventions in their focus on appraisals of a specific stress response, rather than broader beliefs about the nature of stress. Reappraisal interventions targeting test anxiety have been shown to improve test performance on a first college midterm [19], and on the GRE [24], with performance benefits persisting over weeks and months. Might interventions that more broadly target students' stress mindsets similarly improve academic performance, as well as other important outcomes related to well-being?

A recent imagery-based intervention aimed to shift students' valuations of stress to "stress-can-be-enhancing" by encouraging students to think about the positive consequences of stress and what they could do to experience these consequences [9]. The intervention particularly

benefited students with high baseline levels of perceived distress; at a two-week follow-up, these students showed decreases in their perceived distress as well as improved coping, affect, and academic performance. These findings suggest that encouraging college students to shift their valuations of stress may be beneficial, but many questions remain. These questions define the goals of the present study.

**Do stress mindsets vary across sources of stress?** Previously, research has primarily focused on beliefs about *stress in general* as enhancing or debilitating, suggesting that an individual's mindset about the nature of stress is consistent, regardless of the source of that stress. One prior study comparing participants' mindsets about stress in general to their mindsets about stress from their significant current stressor did find that the two stress mindsets were closely related [4]. Yet this finding may be a result of individuals automatically thinking about their most prominent primary stressor when prompted to think about stress in general. The question remains whether beliefs about stress might vary when people think about stress arising from other sources. Given the immense variability in the types of stressors people encounter, and that experiences of stress can vary dramatically depending on the stressor, it may be easier for individuals to adopt a stress-is-enhancing mindset when stress results from certain sources as compared to others. For example, when the stressor is the approaching deadline of a final project, students may be more likely to think of stress as enhancing and supportive of productivity. In contrast, when the stressor is awaiting a grade from an exam you've already taken, it may be harder to view the resulting stress as beneficial.

Stressors vary across at least two dimensions. A first dimension corresponds to duration: stressors may be time-limited (acute) or persist over a longer period (chronic). Stressor duration influences how stress impacts health. Stress in response to acute stressors may be adaptive, but chronic stressors can shift the stress response towards distress. Exposure to chronic stressors is associated with suppressed immunity and predicts poor health outcomes including increased susceptibility to infections and cancer [25], and psychopathology [26].

A second dimension corresponds to controllability: stressors may be perceived as controllable or uncontrollable, a factor which influences how people respond to and cope with the corresponding stress. When people believe they have control over how they react to a stressor, they experience less distress and more stress-related growth [27]. Stress arising from controllable stressors often engenders action and problem-focused coping; in contrast, stress arising from uncontrollable stressors is related to avoidant coping [28]. Perceived stressor controllability may also influence the relationship between stress and health. Rats demonstrate decreased immunocompetence when exposed to an uncontrollable rather than controllable stressor [29]. For humans, viewing an illness as uncontrollable rather than controllable is related to worsened psychological and physical health outcomes [30].

Together, these findings suggest that how debilitating or enhancing stress may be depends, in part, on the source of stress. Stress that arises in response to more chronic and uncontrollable sources tends to be more debilitating than stress that arises from more acute and controllable sources. Do beliefs about the nature of stress likewise vary across these dimensions—are stress mindsets specific to the source of stress (source-specific)? If so, are source-specific mindsets equally predictive of health and performance outcomes? We hypothesized that students would view stress arising from controllable stressors as more enhancing than stress arising from uncontrollable stressors, and would view stress arising from acute stressors as more enhancing than that from chronic stressors.

To our knowledge, no prior research has investigated whether stress mindsets are stable across sources of stress or specific to source duration and controllability. Investigating this question can inform the design of stress-mindset interventions. If certain source-specific mindsets are especially predictive of students' health, well-being, and academic performance,

interventions targeting these source-specific mindsets may be more effective than interventions that target general stress mindsets.

**How does stress mindset impact health?** Research has consistently shown that higher levels of stress are predictive of poorer health [5], but beliefs about stress may moderate this relationship. For example, one study found that stress was only associated with higher rates of mortality when people believed that stress was harmful [6]. Although this study did not look at stress mindsets specifically, its findings support the idea that valuations of stress influence the relationship between stress and health. Thus, in the current study we explored whether stress mindsets, both generally and for different sources of stress, moderated the relationship between measures of stress and measures of health. We predicted a weaker relationship between stress and health for those with a more stress-is-enhancing mindset as compared to those with a more stress-is-debilitating mindset, consistent with the view that holding a more stress-is-enhancing mindset acts as a buffer against the negative impacts of stress.

In addition to buffering against the negative effects of stress on health, stress mindset might also directly improve health by reducing the experience of stress. Recent research with college students has found that believing stress to be enhancing is associated with more adaptive coping, stronger academic performance, and lower perceived stress [9, 11]. Correlational research has found that students who believed stress to be enhancing experienced fewer symptoms of sympathetic nervous system activation, and this in turn predicted better mental and physical well-being, academic performance, and lower perceived stress [11]. In the same study, students who held stress-is-enhancing mindsets were also more likely to engage in more active forms of coping, which was related to psychological well-being and lower perceived stress. Perceived stress as it is measured in this study (and traditionally) functions to measure perceived *distress* (only the negative experiences of stress), and so, these findings suggest that a stress-is-enhancing mindset may benefit students by improving approach coping and reducing distress. The benefit of a stress-is-enhancing mindset on student coping and distress is further supported by a recent imagery-based intervention [9]. Students with high baseline perceived distress benefited from modifying their mindsets to be more enhancing, showing decreased perceived distress and improved coping, as well as improved academic performance, in response to the intervention.

In the current study, we build on this prior work by conducting exploratory analyses to examine approach coping (including planning and problem-solving) and perceived distress as potential mediators of the link between stress mindsets and health. Prior research has found that stress-is-enhancing mindsets are associated with increased use of approach coping [4], and approach coping has been related to better health [31]. Workplace studies suggest that when faced with high stress, individuals with a stress-is-enhancing mindset are more likely to employ approach coping and thus demonstrate improved task performance [32].

We hypothesized that stress-is-enhancing mindsets, both general and source-specific, would be associated with greater use of approach coping and lower perceived distress, which in turn would lead to better mental and physical health. Although the design of our study, with data collected at a single point in time, does not allow firm causal conclusions to be drawn, these exploratory analyses will provide some insight into the potential mechanisms linking general and source-specific stress mindsets to health in college undergraduates.

## The current study

The overarching goal of the current study was to replicate and extend previous research on stress mindsets in a sample of North American college undergraduates drawn from a selective private university and an online task completion marketplace preregistered design and analysis

plan available here: https://osf.io/xrmkn/. Our specific goals were to: (a) expand the current understanding of stress mindsets by exploring whether mindsets vary across sources of stress that vary on dimensions of chronicity and controllability, (b) replicate prior work [11] suggesting that student mindset is related to academic performance and well-being, and (c) evaluate whether mindsets moderate the effects of stress on health. Our final goal was to (d) explore approach coping and perceived stress as potential mediators of the association between stress and health.

## Method

### Design, participants, and procedure

The study employed a cross-sectional, correlational design. Participants were 498 undergraduate students aged 18 to 60 (median age 19–20; 90% of participants were between 18 and 29) from the United States (U.S.) and Canada, recruited from two sources: (a) a psychology participant pool at a mid-sized selective private university in the southeastern U.S., and (b) Amazon Mechanical Turk (MTurk). We recruited as many participants as possible given the available participant pool and funding, and aimed for a sample size that was larger than previously published studies of stress mindset in college students (e.g., [11]; $N$ = 218). After providing consent, participants completed study measures online via Qualtrics; the study took approximately 20 minutes to complete. The study was approved by the Duke University Campus Institutional Review Board, and all data were collected between October and December 2018. The hypotheses and analysis plan for the study were preregistered on the Open Science Framework, available here: https://osf.io/xrmkn/. Details regarding our adherence to the preregistration are also provided in S1 Appendix. To ensure a shorter, more readable manuscript, one preregistered analysis is not presented but is available in the S2 Appendix. This analysis and its results did not affect the substantive conclusions of the current study.

**Psychology participant pool sample.** Psychology pool participants ($n$ = 299; 64.5% women; 47.5% White, 29.0% Asian/Asian American, 12.5% multiracial, 6.4% Black/African American, 3.7% Hispanic/Latino, 1.0% other race/ethnicity) were recruited via an online experiment sign-up program and received course credit for participating. Participants were between 18 and 24 years of age (51.5% freshman, 31.4% sophomores, 9.7% juniors, 7.4% seniors). Consistent with preregistered exclusion criteria, one participant was excluded for nonsense response and for responding to open-ended questions with random strings of letters. The final sample included 298 participant pool participants.

**MTurk sample.** MTurk is an online task-completion system that allows users to complete tasks, including psychological research studies, in exchange for payment or Amazon credit. The study was administered via the online research platform CloudResearch (www.cloudresearch.com). Through CloudResearch's panels, the study was advertised only to potential participants who had previously indicated that they were current college students. Participants were paid $2.20, and a total of 199 participants completed the study (58.7% women; 57.8% White/Caucasian, 14.6% Black/African American, 11.6% Hispanic/Latino, 8.5% multiracial, 6.5% Asian/Asian American, 1.0% other race/ethnicity). Four participants were excluded because they had been out of college for more than two years, consistent with preregistered exclusion criteria. The final sample included 195 MTurk participants (18 to 60 years of age; median age 24; 7.0% freshman, 26.9% sophomores, 25.8% juniors, 40.3% seniors).

### Measures

**General stress mindsets.** A revised 3-item version of the 8-item General Stress Mindset scale [4] was administered, assessing participants' views of stress as enhancing versus

debilitating. Three rather than eight items were selected for the current study to reduce survey length, given that participants would be responding to the items for stress generally and also for specific sources of stress (described in the following section). Participants responded to the three items (i.e., "Experiencing stress facilitates my learning and growth," "Experiencing stress debilitates my performance and productivity" [reverse-scored], "The effects of stress are positive and should be utilized") on a 5-point scale (0 = *strongly disagree*, 4 = *strongly agree*). Higher scores indicate a more stress-is-enhancing mindset.

We were able to compare the internal reliability of the full 8-item scale to that of the briefer 3-item scale using data from an independent sample collected by the third author. Participants (n = 1042) were college undergraduates enrolled in an introductory psychology course at the same university from which the participant pool sample was drawn, who completed the 8-item Stress Mindset Scale [4] for course credit across five semesters between fall 2018 and spring 2020. In this independent sample, internal reliability for the full 8-item scale was .84, and for our briefer 3-item scale was .71. The correlation between the 8-item scale and the 3-item scale was .91, p < .001, indicating that this briefer measure is very strongly correlated with the full 8-item scale.

**Source-specific stress mindsets.** To assess beliefs about stress with regard to specific types of academic sources of stress, participants were asked to consider a potentially stressful experience that was either acute or chronic, and controllable or uncontrollable. They were given a definition for the terms acute/chronic and controllable/uncontrollable (e.g., "One type of potentially stressful experience is acute and uncontrollable. This is a short-term experience which you cannot prepare for or do anything about.") followed by an example. The examples consisted of one of four hypothetical stressful academic situations designed to vary in duration (more short-term, or acute, stressors, versus more long-term, or chronic, stressors) and controllability (more controllable versus more uncontrollable stressors). To avoid introducing additional sources of variability, and to ensure that situations would be relevant to undergraduate students from a variety of backgrounds, all of the hypothetical stressful situations were in the academic domain. Table 1 includes the full text of each situation. The hypothetical situations were generated based on a focus group discussion conducted in spring 2018 with eight undergraduate students at the same university as the participant pool sample (a report of this focus group discussion is available from the authors on request). Participants were asked to think about the stress generated by each situation and were asked about their mindset for that

**Table 1. Specific stress mindset hypothetical situations for acute controllable, chronic controllable, acute uncontrollable, and chronic uncontrollable mindset.**

|  | Controllable | Uncontrollable |
|---|---|---|
| **Acute** | One type of potentially stressful experience is *acute* and *controllable*. This is a short-term experience for which you can in some way prepare. | One type of potentially stressful experience is *acute* and *uncontrollable*. This is a short-term experience which you cannot prepare for or do anything about. |
|  | For example, imagine having to give a class presentation that you are really dreading. You may experience stress over the days leading up to the presentation. However, you know the date of the presentation, and you know what the requirements are to do well. | For example, imagine waiting to find out how you did on an important exam/assignment. You may feel stressed awaiting your score, which should arrive in just a few days. There is nothing you can do at this point to change your score or to determine when it will be released; all you can do is wait. |
| **Chronic** | One type of potentially stressful experience is *chronic* and *controllable*. This is a long-term experience for which you can in some way prepare. | One type of potentially stressful experience is *chronic* and *uncontrollable*. This is a long-term experience for which you cannot prepare. |
|  | For example, imagine taking a class that has a difficult quiz at the beginning of every class period. You may experience stress about these quizzes each week, all semester. The quizzes are challenging, but you know what they will cover. | For example, imagine taking a class in which the instructor practices the Socratic method (where students are called on by the professor and asked a series of difficult, probing questions to assess their understanding and assumptions). You may experience stress fearing you will be called on each class, all semester. You have no idea when you will be called on, and because of the difficult and unpredictable nature of the questions it is difficult to adequately prepare beforehand. |

kind of stress. Participants responded to the three items administered in the general stress mindsets measure, revised with the term "this kind of stress" in place of "stress."

To examine the factor structure of the adapted 3-item General Stress Mindset Scale and the newly developed Source-specific Stress Mindsets Scales, a confirmatory factor analysis was conducted to evaluate the fit of a baseline measurement model with general and stressor-specific mindsets items loading onto five correlated first-order factors (see S3 Appendix for CFA details, including items, factor loadings, and path diagram). Fit statistics, with the exception of the SRMR statistic which exceeds the suggested cutoff, indicated that the hypothesized model presented an adequate fit to the data $\chi^2$ (75) = 205.49, $p$ < .001; RMSEA = .059 [90% CI .050–.069]; CFI = .948; SRMR = .113.

**Perceived stressfulness of hypothetical stressful situations.** After responding to the specific stress mindsets items, participants were presented again with each hypothetical stressful situation and asked to indicate on a 5-point scale how stressful they perceived each situation to be (1 = *not at all stressful*, 5 = *extremely stressful*). This stressfulness rating allowed for comparison of how distressing participants found these hypothetical situations compared to one another, and to their own primary sources of stress (collected as part of the Brief COPE questionnaire).

**Self-reported health.** The Health-Related Quality of Life Scale (HRQOL) is a four-item measure developed by the Centers for Disease Control and Prevention [33] and validated in college populations [34]. In the current study, we use the item "Would you say that in general your health is excellent, very good, good, fair, or poor?" as an indicator of perceptions of overall health (with higher scores indicating *poorer* perceived health); and the item "During the past 30 days, for about how many days did poor physical or mental health keep you from doing your usual activities, such as self-care, work, or recreation?" as an indicator of how often poor physical and mental health interfered with normal activities (higher scores indicate more days on which health interfered with normal activities). Note that number of days health interfered with normal activities had missing data for 59 participants, we expect due to a survey design flaw. Participants responded to this item using a slider response format, which was set to 0 by default, but participants needed to click on the slider for their response to be recorded as 0. We expect that some or all of the 59 participants with missing data on this measure may have intended to record a response of 0 but did not click on the slider for their response to be recorded. To examine whether missing data may have affected study results, we re-ran all analyses involving this measure assigning all participants with missing data a response of 0. This changed the specific point estimates for some statistics, but did not change any substantive conclusions. The results presented in the paper are presented with the 59 participants with missing data excluded, and results including the 59 participants with responses of 0 are available from the authors upon request.

**Mental health symptoms.** To assess symptoms of mental health problems, participants completed three subscales of the Symptom Checklist–90 [35], assessing how much symptoms of depression (12 items), anxiety (10 items), and somatization (12 items) had bothered them in the past month (0 = *not at all*, 4 = *extremely*). Internal reliability was high for depression (α = .93), anxiety (α = .93), and somatic symptoms (α = .92). For the purposes of data reduction, a single composite mental health symptoms score was created that was the average of the mean score for each of the subscales, with higher scores indicating higher levels of depression, anxiety, and somatic symptoms; the composite was derived from the subscale mean scores to avoid weighting subscales with more items more heavily in the overall composite. The internal reliability of this composite was high (α = .96).

**Number of experienced stressful life events.** To measure experience of stressful life events, we used an adapted version of the Life Events Scale for Students (LESS) [36, 37], which

asks participants whether they have ever experienced a series of potentially stressful events. Several events from the original measure were excluded due to lack of relevance for this college sample (e.g., vacation alone/with friends), and additional events were included that were considered relevant (e.g., experiencing a natural disaster). The adapted version of the measure included 28 events. Each item was weighted on a scale from 0 (not at all stressful) to 100 (extremely stressful) based on ratings from undergraduates made in previous research [38, 39]. New events added to the measure were weighted by the authors with reference to existing weights. A weighted sum score was created for each participant, with higher scores indicating more experienced stressful life events. Findings were similar using the weighted and unweighted versions of the sum score.

**Perceived distress.** To assess current levels of experienced distress, participants responded to the 10-item Perceived Stress Scale [14, 40]. Participants responded to items (e.g., "In the past month, how often have you felt nervous and 'stressed'?", "In the past month, how often have you felt that you were unable to control the important things in your life?") on a 5-point scale (0 = *never*, 4 = *very often*) indicating how often they thought or felt a certain way over the past month (α = .86). A sum score was created, with higher scores indicating greater perceived distress.

**Coping.** Participants completed a 26-item version of the Brief COPE [38], assessing 13 potential responses to stressful events (i.e., active coping [Spearman-Brown coefficient = .77], planning [S-B = .74], positive reframing [S-B = .76], acceptance [S-B = .49], humor [S-B = .85], religion [S-B = .89], seeking emotional support [S-B = .84], seeking instrumental support [S-B = .85], self-distraction [S-B = .54], denial [S-B = .72], venting [S-B = .67], behavioral disengagement [S-B = .65], self-blame [S-B = .81]). Items related to substance use were excluded to avoid asking about illegal behaviors in participants who were underage.

Participants indicated the current most stressful thing in their lives, rated how stressful they found it (1 = *not stressful at all*, 7 = *extremely stressful*), and then rated how much they were using each coping strategy in response to that source of stress (1 = *I haven't been doing this at all*, 5 = *I've been doing this a lot*). To reduce the number of coping scales, we created four composites reflecting approach coping (active coping, planning, positive reframing, acceptance; α = .75), social coping (seeking emotional support, seeking instrumental support, venting; α = .83), distractive coping (humor, religion, self-distraction; α = .52), and avoidant coping (denial, behavioral disengagement, self-blame; α = .75), following the approach taken by Crum et al. [4], see pp. 719–721 for validity information.

**Academic performance.** As a measure of academic performance, students were asked to self-report their cumulative undergraduate grade point average (GPA) on a four-point scale, with higher scores indicating higher GPA. Because data were collected before the end of the first semester of the academic year, first-year students were asked to estimate GPA to the best of their ability. For analyses including GPA, analyses were conducted both with and without first-year students because their reports may have been more inaccurate as at the time of the survey freshman had not yet completed a full semester of college.

## Results

Zero-order correlations among study variables are presented in Table 2. Many study variables, including stress mindsets, perceived distress, history of stressful life events, mental and physical health, and GPA, varied as a function of sample (psychology participant pool, MTurk) and, to a lesser extent, gender (see S1 and S2 Tables for means, SDs, *F* ratios, and effect sizes). Thus, sample and gender were included as control variables in all subsequent analyses. Partial correlations, partialling out variance associated with sample and gender, are also included in Table 2, above the diagonal.

Table 2. Zero-order and partial (controlling for gender and sample) correlations among study variables.

| | 1 | 2 | 3 | 4 | 5 | 6 | 7 | 8 | 9 | 10 | 11 | 12 | 13 | 14 | 15 | 16 | 17 |
|---|---|---|---|---|---|---|---|---|---|---|---|---|---|---|---|---|---|
| 1. Gender | — | — | — | — | — | — | — | — | — | — | — | — | — | — | — | — | — |
| 2. Sample | -.06 | — | — | — | — | — | — | — | — | — | — | — | — | — | — | — | — |
| 3. SM-G | -.11* | -.30*** | — | .37*** | .34*** | .21*** | .28*** | -.22*** | -.02 | -.19*** | -.11* | -.07 | .20*** | -.01 | .05 | -.08 | .09 |
| 4. SMS-AC | .03 | -.09 | .33*** | — | .29*** | .01 | .09 | -.14* | .03 | -.14** | -.06 | -.09 | .20*** | .04 | .04 | -.08 | -.04 |
| 5. SMS-CC | -.10* | -.04 | .33*** | .25*** | — | .15** | .19*** | -.23*** | -.05 | -.22*** | -.15** | -.18* | .21*** | -.01 | -.01 | -.21*** | -.04 |
| 6. SMS-AU | .01 | .23*** | .09 | .01 | .11* | — | .05 | -.17*** | .06 | -.14** | -.12* | -.02 | .20*** | -.04 | .05 | .00 | -.03 |
| 7. SMS-CU | -.15** | -.03 | .27** | .05 | .19*** | .05 | — | -.07 | .04 | -.04 | -.04 | .05 | .15*** | .15*** | .11* | .04 | .01 |
| 8. Perceived Stress | .14** | .12** | -.24*** | -.16** | -.23*** | -.14** | -.08 | — | .26*** | .70*** | .50*** | .46*** | -.14** | .15** | .18*** | .59*** | -.20*** |
| 9. Stressful Life Events-Weighted | .01 | .45*** | -.16*** | -.02 | -.06 | .14** | .01 | .30*** | — | .34*** | .26*** | .26*** | .12* | .07 | .13** | .13** | -.11* |
| 10. Mental Health Composite | -.18** | -.21*** | -.25*** | -.17** | -.24*** | -.10* | -.08 | .72*** | .40*** | — | .39*** | .50*** | -.12* | .12* | .19*** | .51*** | -.11* |
| 11. Self-Reported Health | .10* | .22*** | -.18*** | -.09 | -.16*** | -.08 | -.05 | .52*** | .32*** | .46*** | — | .41*** | -.10* | .05 | .03 | .28*** | -.09 |
| 12. Days Health Interfered | .13** | .14** | -.13** | -.12* | -.20*** | .00 | .01 | .48*** | .31*** | -.54*** | .43*** | — | -.05 | .15** | .19*** | .36*** | -.10 |
| 13. Approach Coping | -.04 | .02 | .19*** | .20*** | .20*** | .18*** | .13** | -.15** | .10* | -.12* | -.11* | -.05 | — | .35*** | .18*** | -.14** | -.08 |
| 14. Social Coping | .16** | -.10 | -.02 | .03 | -.02 | -.06 | .12** | .16*** | .04 | .14** | .04 | .32*** | .35*** | — | .35*** | .23** | -.04 |
| 15. Distractive Coping | .05 | -.04 | .06 | .02 | -.05 | .03 | .01** | .16** | .09* | .16*** | .01 | .17** | .16*** | .35*** | — | .36*** | -.14** |
| 16. Avoidant Coping | .12* | .05 | -.10* | -.09* | -.22** | -.01 | .04 | .59*** | .17*** | .52*** | .31*** | .37*** | -.15** | .23*** | .34*** | — | -.20*** |
| 17. GPA | .12* | -.31*** | .14** | -.01 | -.03 | -.12** | .01 | -.19*** | -.22*** | -.14** | -.14** | -.11* | -.10* | .01 | -.10* | -.19*** | — |

*Note*. Correlations presented below the diagonal are zero-order correlations; correlations presented above the diagonal are partial correlations partialling out gender (0 = *men*, 1 = *women*) and sample (0 = *psychology participant pool*, 1 = *Amazon Mechanical Turk*). Table abbreviations are as follows: SM-G = Stress Mindsets Scale-General, SMS-AC = Specific Stress Mindsets Scale–Acute Controllable situation, SMS-CC = Specific Stress Mindsets Scale–Chronic Controllable situation, SMS-AU = Specific Stress Mindsets Scale–Acute Uncontrollable situation' SMS-CU = Specific Stress Mindsets Scale–Chronic Uncontrollable situation * $p < .05$, ** $p < .01$, *** $p < .001$.

## General versus source-specific stress mindsets

A main goal of this study was to examine how stress mindsets might vary across different sources of stress. First, we needed to examine the psychometric properties of the newly developed source-specific stress mindsets measures and how source-specific mindsets relate to beliefs about stress in general as enhancing vs. debilitating. Internal reliability for the three-item general stress mindset scale was adequate ($\alpha$ = .69), if somewhat lower than a three-item version of the measure recently developed for use with adolescents ($\alpha$ = .77, [39]). With regard to the source-specific stress mindsets scales, internal reliability was generally stronger and more comparable to the Crum et al. [4] and Park et al. [39] measures (acute controllable $\alpha$ = .76, chronic controllable $\alpha$ = .79, acute uncontrollable $\alpha$ = .70, chronic uncontrollable $\alpha$ = .83).

Correlations among the general and source-specific stress mindsets measures were very modest (*r*s ranging from .01 to .33, see Table 2), and in some cases non-significant, indicating that participants' mindsets did vary as a function of the features of sources of stress, and that views about stress in general as enhancing versus debilitating do not necessarily predict views about stress in response to specific stressful situations.

Using the stressfulness ratings participants provided for each of the hypothetical stressful situations, we examined whether mindsets were associated with the perceived stressfulness of each situation (see Table 3). Participants rated the hypothetical situations as less stressful than their primary stressor; the chronic controllable situation was rated as most stressful and the chronic uncontrollable as least. We also examined the association between participants'

**Table 3. Correlations among stress mindsets and stressfulness ratings.**

|  | 1 | 2 | 3 | 4 | 5 | 6 | 7 | 8 | 9 | 10 |
|---|---|---|---|---|---|---|---|---|---|---|
| Mean (SD) | 1.90 (.81) | 2.37 (.83) | 2.42 (.87) | 1.39 (.80) | 1.65 (1.03) | 5.79 (1.14) | 3.41 (1.08) | 3.09 (1.09) | 3.19 (1.20) | 4.06 (1.05) |
| 1. General Stress Mindset | — |  |  |  |  |  |  |  |  |  |
| 2. Specific Stress Mindset–Acute Controllable | .33*** | — |  |  |  |  |  |  |  |  |
| 3. Specific Stress Mindset–Chronic Controllable | .33*** | .25*** | — |  |  |  |  |  |  |  |
| 4. Specific Stress Mindset–Acute Uncontrollable | .09 | .01 | .11* | — |  |  |  |  |  |  |
| 5. Specific Stress Mindset–Chronic Uncontrollable | .27*** | .05 | .19*** | .05 | — |  |  |  |  |  |
| 6. Stressfulness Rating–Primary Stressor | -.12** | -.06 | -.13** | -.04 | .00 | — |  |  |  |  |
| 7. Stressfulness Rating–Acute Controllable | -.08 | -.28*** | -.07 | .01 | -.07 | .17*** | — |  |  |  |
| 8. Stressfulness Rating–Chronic Controllable | -.09 | -.04 | -.32*** | .01 | -.10* | .23*** | .10* | — |  |  |
| 9. Stressfulness Rating–Acute Uncontrollable | -.05 | -.02 | -.05 | -.22*** | .05 | .21*** | -.03 | .07 | — |  |
| 10. Stressfulness Rating–Chronic Uncontrollable | -.14** | .05 | -.05 | .01 | -.46 *** | .08 | .25*** | .19*** | .01 | — |

*Note.* Stressfulness ratings are on a 5-point scale (1 = *not at all stressful*, 5 = *extremely stressful*); for the primary stressor, participants listed the current most stressful thing in their life and rated its' stressfulness; for the specific stress mindsets situations, participants rated the perceived stressfulness of each situation after they had provided all the mindset ratings.

* $p < .05$

** $p < .01$

*** $p < .001$.

stressfulness rating for their primary stressor and their general stress mindset. Overall, mindsets and stressfulness ratings were moderately negatively correlated, indicating that participants viewed stress as more enhancing in situations that they perceived to be less stressful. General and chronic controllable stress mindsets were related to how stressful participants reported their primary stressors, again with participants who endorsed more of a stress-is-enhancing mindset viewing their primary stressor as less stressful.

## Views of stress as enhancing versus debilitating

The next step in our analyses was to examine students' overall views of stress as enhancing versus debilitating, and to examine whether views of stress varied as a function of the duration (acute versus chronic) and controllability (controllable versus uncontrollable) of the stressor (see S2 Table for overall sample means for general and source-specific stress mindset measures). With regard to the first question, consistent with previous research [4, 11, 42], students in our sample viewed stress in general as more debilitating than enhancing (M = 1.90, SD = 0.81, on a 0 to 4 scale).

To examine whether stress mindsets varied as a function of stressor type, a multivariate analysis of variance (MANOVA) was conducted with controllability (controllable versus uncontrollable) and duration of the stressor (acute versus chronic) as within-subjects factors, and gender and sample as between-subjects factors. Consistent with expectations, participants rated stress arising from uncontrollable stressors (M = 1.52, SD = 0.67) as significantly more debilitating than stress arising from controllable stressors (M = 2.40, SD = 0.67), Wilks' λ = .531, $F(1, 481) = 424.96$, $p < .001$, partial $\eta^2 = .469$. However, in contrast to expectations, participants rated stress arising from acute stressors (M = 1.89, SD = 0.59) as more debilitating than stress from chronic stressors (M = 2.04, SD = 0.74), Wilks' λ = .961, $F(1, 481) = 19.34$, $p <$

.001, partial $\eta^2$ = .039. The effect of controllability was much larger than the small effect of duration, as indicated by the larger partial $\eta^2$ value.

The main effects of controllability and duration were qualified by a significant multivariate interaction with a small effect size (Wilks' $\lambda$ = .989, $F(1, 481)$ = 5.47, $p$ = .020, partial $\eta^2$ = .011). Simple effects tests indicated that stress from acute stressors was perceived as more debilitating than stress from chronic stressors specifically when the stressor was also uncontrollable (Wilks' $\lambda$ = .961, $F(1, 481)$ = 19.33, $p < .001$, partial $\eta^2$ = .039), but not when it was controllable (Wilks' $\lambda$ = .994, $F(1, 483)$ = 2.88, $p$ = .091, partial $\eta^2$ = .006). This interaction was driven by the rating of stress from the acute uncontrollable situation (waiting for a test grade after you have completed the test) as particularly debilitating ($M$ = 1.39, $SD$ = .80).

## Links between stress, stress mindsets, physical and mental health, and academic performance

Consistent with prior research [40], zero-order and partial correlations showed significant and sizeable associations between amount of stress (life events and perceived distress) and poorer physical and mental health (see Table 2). Also consistent with previous research [4, 11], stress-is-debilitating mindsets were related to higher levels of perceived distress and to poorer health. As expected, participants who had experienced more lifetime stress also reported poorer mental and physical health, even controlling for sample and gender. There were no significant associations between any of the general or specific stress mindset measures and history of stressful life events when controlling for sample and gender.

With regard to GPA, we hypothesized, based on previous research [11], that students who viewed stress as more enhancing would also report stronger academic performance. Controlling for sample and gender, the relationship between mindset and GPA was no longer statistically significant for general mindsets ($r$ = .09, $p$ = .056) or for specific stress mindsets. When first-year students were excluded from the analyses to remove possible unreliable reports of GPA, the relationship between mindset and GPA remained non-significant for general mindsets ($r$ = .10, $p$ = .096) and source-specific mindsets ($r$s ranging from -.08 for acute uncontrollable stress mindset to .15 for chronic uncontrollable mindsets), controlling for sample and gender.

## Stress mindsets as moderators in the association between stress exposure and health

Correlational analyses indicate that levels of stress and viewing stress as more debilitating are each associated with poorer health. This next set of analyses examined whether stress mindsets moderate the link between stress and physical and mental health. Our hypothesis was that viewing stress as enhancing would buffer the negative relationship between stress and health. To address these questions, we performed several sets of four-step hierarchical multiple regression analyses. In each set of analyses: (a) gender (0 = *men*, 1 = *women*) and sample (0 = *participant pool*, 1 = *MTurk*) were entered as control variables in the first step of the model, (b) the two mean-centered stress variables (history of stressful life events [weighted sum] and perceived distress) were entered in the second step of the model, (c) the mean-centered stress mindset variable was entered in the third step of the model, and (d) interaction terms between the stress mindset and each of the stress variables were included in the fourth step of the model. To build a thorough understanding of the relationship between stress mindset and health, we estimated 15 separate models: one set of models for general stress mindset and one set for each of the four source-specific stress mindsets, separately for each of the three health measures (mental health symptoms, general self-reported poor health, and number of days health interfered with normal activities). Results are summarized in Table 4.

**Table 4. Hierarchical multiple regression analyses predicting health measures from history of stressful life events, perceived stress, and general stress mindset.**

| Predictor | Mental Health Symptoms | | | General Health | | | Number of Days Health Interfered with Activities | | |
|---|---|---|---|---|---|---|---|---|---|
| | $\Delta R^2$ | $b$ (se) | $sr^2$ | $\Delta R^2$ | $b$ (se) | $sr^2$ | $\Delta R^2$ | $b$ (se) | $sr^2$ |
| Step 1 | .076*** | | | .056*** | | | .034** | | |
| Control variables[a] | | | | | | | | | |
| Step 2 | .483*** | | | .249*** | | | .224*** | | |
| Stressful life events (weighted) | | .0005 (.00009)*** | .024 | | .0004 (.0001)** | .014 | | .003 (.001)** | .020 |
| Perceived stress | | -.07 (.00)*** | .376 | | .06 (.01)*** | .190 | | .37 (.04)*** | .158 |
| **Step 3A and 4A: General Stress Mindset** | | | | | | | | | |
| Step 3A | .002 | | | .000 | | | .000 | | |
| Stress mindset (general) | | -.05 (.03) | .002 | | -.03 (.05) | .000 | | .18 (.36) | .000 |
| Step 4A | .07* | | | .009* | | | .016** | | |
| Life events x mindset | | .0002 (.0001) | .002 | | .00008 (.0002) | .000 | | -.003 (.001)** | .015 |
| Perceived stress x mindset | | -.01 (.00)** | .007 | | .01 (.01)* | .007 | | .08 (.05) | .005 |
| Adjusted $R^2$ | | .562*** | | | .304*** | | | .263*** | |
| **Step 3B and 4B: Specific Stress Mindset–Acute Controllable** | | | | | | | | | |
| Step 3B | .003 | | | .000 | | | .002 | | |
| Stress mindset (acute controllable) | | -.06 (.03)* | .003 | | .01 (.05) | .000 | | -.34 (.33) | .002 |
| Step 4B | .000 | | | .000 | | | .003 | | |
| Life events x mindset | | -.00002 (.0001) | .000 | | .00002 (.0002) | .000 | | -.001 (.001) | .002 |
| Perceived stress x mindset | | .00 (.00) | .000 | | .00 (.01) | .000 | | .04 (.04) | .002 |
| Adjusted $R^2$ | | .556*** | | | .297*** | | | .251*** | |
| **Step 3C and 4C: Specific Stress Mindset–Acute Uncontrollable** | | | | | | | | | |
| Step 3C | .003 | | | .004 | | | .001 | | |
| Stress mindset (acute uncontrollable) | | -.06 (.03) | .003 | | -.08 (.05) | .004 | | .31 (.35) | .001 |
| Step 4C | .009** | | | .002 | | | .009 | | |
| Life events x mindset | | .0001 (.0001) | .001 | | .00007 (.0002) | .000 | | -.002 (.001)* | .009 |
| Perceived stress x mindset | | -.01 (.00)** | .009 | | -.01 (.01) | .002 | | .02 (.04) | .000 |
| Adjusted $R^2$ | | .565*** | | | .302*** | | | .257*** | |
| **Step 3D and 4D: Specific Stress Mindset–Chronic Controllable** | | | | | | | | | |
| Step 3D | .004* | | | .002 | | | .008* | | |
| Stress mindset (chronic controllable) | | -.06 (.03)* | .004 | | -.05 (.05) | .002 | | -.64 (.31)* | .007 |
| Step 4D | .001 | | | .003 | | | .022** | | |
| Life events x mindset | | -.00006 (.0001) | .000 | | .0002 (.0001) | .002 | | -.003 (.001)** | .019 |
| Perceived stress x mindset | | .00 (.00) | .000 | | .00 (.01) | .000 | | -.03 (.04) | .000 |
| Adjusted $R^2$ | | .558*** | | | .297*** | | | .275*** | |
| **Step 3E and 4E: Specific Stress Mindset–Chronic Uncontrollable** | | | | | | | | | |
| Step 3E | .000 | | | .000 | | | .004 | | |
| Stress mindset (chronic uncontrollable) | | -.01 (.02) | .000 | | -.02(.04) | .000 | | .37 (.27) | .004 |
| Step 4E | .000 | | | .003 | | | .006 | | |
| Life events x mindset | | .00005 (.00008) | .000 | | .0002 (.0001) | .003 | | -.0005 (.001) | .000 |
| Perceived stress x mindset | | .00 (.00) | .000 | | .00 (.01) | .000 | | -.07 (.04) | .006 |
| Adjusted $R^2$ | | .554*** | | | .300*** | | | .255*** | |

*Note.* Each column represents a separate hierarchical multiple regression analysis, with the dependent variable listed in the column header. [a] control variables are gender (0 = men, 1 = women) and sample (0 = psychology participant pool, 1 = Amazon Mechanical Turk). $sr^2$ = squared semi-partial correlation.

* $p < .05$

** $p < .01$

*** $p < .001$.

**Table 5. Simple intercepts and simple slopes for significant interaction effects between stress measures and stress mindsets predicting health.**

| | More Debilitating Mindset (20th percentile) | | Average Mindset (50th percentile) | | More Enhancing Mindset (80th percentile) | |
|---|---|---|---|---|---|---|
| | Simple Intercept | Simple Slope | Simple Intercept | Simple Slope | Simple Intercept | Simple Slope |
| *Predicting Mental Health Symptoms from Perceived Stress* | | | | | | |
| Moderated by General Mindset | 1.12 (.06) | .08 (.005) | 1.07 (.04) | .07 (.003) | 1.04 (.05) | .06 (.01) |
| Moderated by Acute Uncontrollable Mindset | 1.06 (.05) | .08 (.004) | 1.01 (.04) | .07 (.004) | .96 (.05) | .06 (.004) |
| *Predicting General Self-Reported Health from Perceived Stress* | | | | | | |
| Moderated by General Mindset | 2.40 (.10) | .05 (.01) | 2.37 (.08) | .06 (.01) | 2.36 (.08) | .07 (.01) |
| *Predicting Days Health Interfered with Normal Activity from History of Stressful Life Events* | | | | | | |
| Moderated by General Mindset | 4.94 (.68) | .01 (.001) | 5.20 (.53) | .003 (.001) | 5.37 (.55) | .00 (.00)[ns] |
| Moderated by Acute Uncontrollable Mindset | 5.34 (.56) | .01 (.001) | 5.62 (.53) | .004 (.001) | 5.90 (.60) | .003 (.001) |
| Moderated by Chronic Controllable Mindset | 5.67 (.56) | .005 (.001) | 5.25 (.52) | .002 (.001) | 5.11 (.53) | .001 (.001)[ns] |

*Note.* The superscript *ns* indicates that the simple slope was not significantly different from zero; all other simple intercepts and simple slopes are significantly different from zero at $p < .05$.

With regard to main effects, history of stressful life events and perceived distress were each uniquely related to health measures after controlling for sample and gender. The two stress measures explained a significant proportion of variance in each outcome—48.3% of the variance in mental health symptoms, 24.9% of the variance in general self-reported poor health, and 22.4% of the variance in number of days health interfered with normal activities (Table 4). For stress mindsets, however, there were no significant main effects of general stress mindsets, acute controllable stress mindsets, acute uncontrollable stress mindsets, or chronic uncontrollable mindsets once sample, gender, and stress variables were taken into account. The only mindset for which there were significant main effects was the chronic controllable mindset, such that viewing stress as more enhancing in that situation was related to fewer days on which health interfered with normal activities and lower levels of mental health symptoms.

With regard to interaction effects, there was evidence that mindsets moderated the association between stress and health for some mindsets, but not others. When significant interaction effects were observed, they were probed using the Preacher, Curran, and Bauer [41] online interaction utility. Simple slopes were calculated for participants at the 20th, 50th, and 80th percentile on the mindset variables. For ease of interpretation, a summary of model results is presented in Table 5, with a focus on statistically significant interactions.

With respect to mental health symptoms, the relationship between stressful life events and health did not vary as a function of mindsets, but the relationship between perceived distress and health did. The effect of perceived distress on mental health varied as a function of general and acute uncontrollable stress mindsets (see Fig 1). In each case of moderation, the effect of perceived distress on health was significant at each level of mindset, but the effect was stronger for participants at the 20th percentile (i.e., for participants with more debilitating mindsets). However, enhancing mindsets did not fully buffer the effect of perceived distress on mental health.

With respect to general self-reported poor health, the effect of stressful life events on self-reported health did not vary as a function of mindset, but the effect of perceived distress did vary as a function of mindset, for general stress mindsets *only* (Fig 2). The effect of perceived distress on self-reported health was significant at each level of mindset, but, in contrast to other measures of health, the effect was strongest at the 80th percentile (i.e., for participants with more enhancing mindsets). That is, perceived distress was more strongly related to general self-reported health for those with more enhancing mindsets.

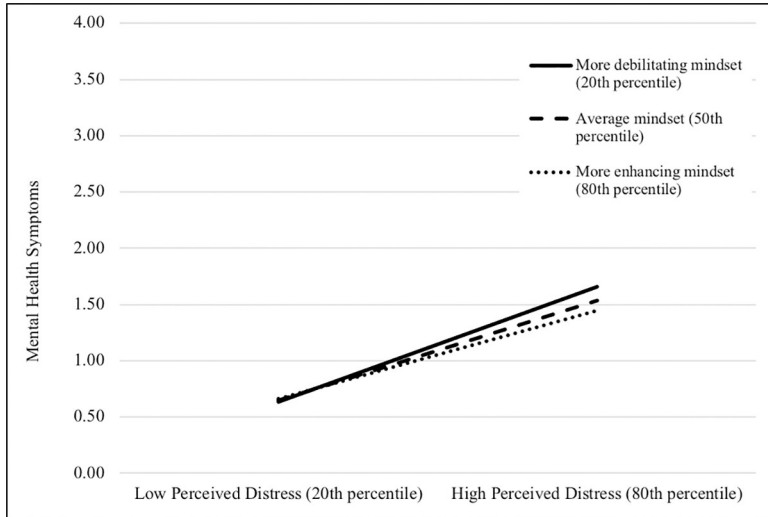

**Fig 1. Graphical representation of interaction between perceived distress and general stress mindset predicting mental health symptoms.**

Finally, with respect to number of days health interfered with normal activity, there was no evidence of moderation for the link between perceived distress and number of days health interfered with normal activity. Mindset did, however, act as a moderator in the link between history of stressful life events and number of days health interfered with normal activity for general, acute uncontrollable, and chronic controllable stress mindsets. Stress-is-enhancing mindsets again acted to partially buffer participants from the effect of a history of stressful life events. The effect was significant for all levels of mindsets for chronic controllable mindsets, and significant for participants in the 20th and 50th percentile for general and acute uncontrollable mindsets (see Fig 3 for illustrative results).

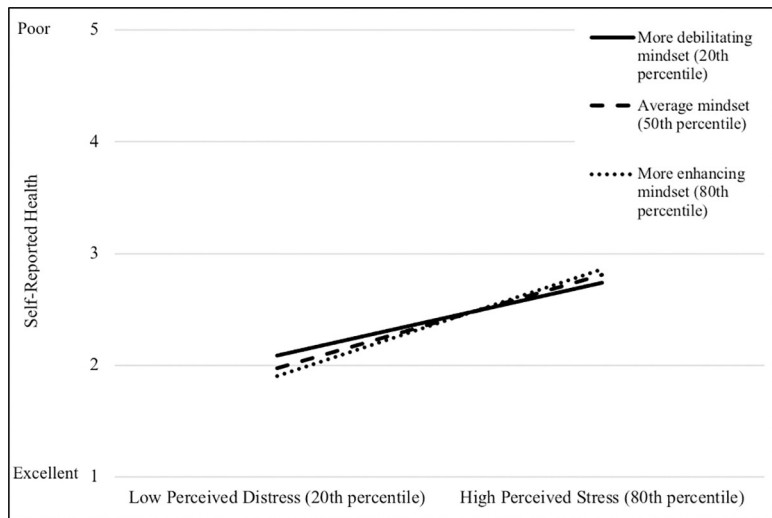

**Fig 2. Graphical representation of interaction between perceived distress and general stress mindset predicting general self-reported health.**

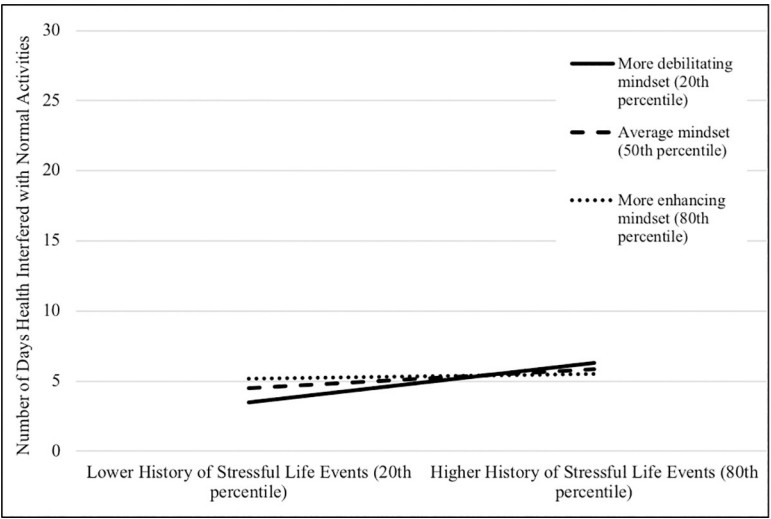

**Fig 3. Graphical representation of interaction between history of stressful life events and general stress mindset predicting days health interfered with normal activity.**

## Exploratory analyses: Approach coping and perceived distress as potential mediators of the link between stress mindset and mental and physical health

In the previous analysis, we found that stress mindset moderated the relationship between stress and health, meaning that stress and health were more weakly connected for those who reported seeing stress as more enhancing. In that analysis, stress was operationalized based on history of stressful life events, but also by a measure of perceived distress that has commonly been used as a stress index [14, 22].

In addition to buffering against the negative effects of perceived distress on health, stress mindset might also directly improve health by *reducing* perceived distress, specifically by increasing the use of approach coping strategies. The next analysis tested this possibility. We conducted a set of serial mediation analyses to calculate indirect effects of mindsets on health through approach coping and perceived distress using Hayes's PROCESS macro for SPSS [42] (see S1 Fig for the hypothesized process model). For each measure of stress mindset, a set of three models was estimated, estimating the indirect effect of mindset on (a) symptoms of mental health problems, (b) self-reported poor health, and (c) number of days health interfered with normal activity in the past 30 days, through approach coping and perceived distress. The hypothesized causal pathway is captured by the estimate of the specific indirect effect of mindset on health through approach coping and perceived distress. Indirect effect estimates were calculated with 95% bias-corrected bootstrap confidence intervals (with 10,000 resamples), and all models were estimated statistically controlling for sample, gender, and previous history of stressful life events. It is important to emphasize that data for this study were collected at a single point in time and so the data cannot provide direct evidence of the hypothesized causal pathway linking stress mindsets to health through approach coping and perceived distress, but rather whether the data are consistent or inconsistent with the hypothesized process.

Indirect effect estimates for the serial effect of mindset on health through approach coping and perceived distress are presented in Table 6, and a complete list of indirect effects estimates for each model are presented in S3–S7 Tables. The specific indirect effect of stress mindset on health through approach coping and perceived distress, although small, was significantly

**Table 6. Specific indirect effects of stress mindsets on health through approach coping and perceived stress.**

| | Mental Health Symptoms | | General Self-Reported Poor Health | | Number of Days Health Interfered with Normal Activity | |
| | Point Estimate (95% CI) | $R^2$ | Point Estimate (95% CI) | $R^2$ | Point Estimate (95% CI) | $R^2$ |
|---|---|---|---|---|---|---|
| *Stress Mindset* | | | | | | |
| General | -.018 (-.036, -.005) | .563 | -.016 (-.031, -.004) | .309 | -.093 (-.188, -.021) | .259 |
| Acute Controllable | -.017 (-.035, -.005) | .564 | -.015 (-.030, -.004) | .311 | -.094 (-.193, -.023) | .260 |
| Chronic Controllable | -.016 (-.033, -.004) | .565 | -.014 (-.028, -.003) | .309 | -.085 (-.175, -.016) | .266 |
| Acute Uncontrollable | -.015 (-.032, -.004) | .564 | -.013 (-.027, -.003) | .313 | -.094 (-.193, -.019) | .260 |
| Chronic Uncontrollable | -.011 (-.024, -.002) | .562 | -.009 (-.020, -.002) | .309 | -.065 (-.138, -.014) | .262 |

*Note.* Confidence intervals are 95% bias-corrected bootstrap confidence intervals with 10,000 resamples calculated using Hayes's PROCESS macro for SPSS (v. 26.0) [42]; indirect effects are significantly different from zero when the associated confidence interval does not contain zero. Point estimates are from serial mediation models including mindset as the predictor, approach coping and perceived stress as serial mediators (in that order), and the health and academic performance measures as outcomes (see S1 Fig); all analyses are controlling for sample, gender, and history of stressful life events; $R^2$ values are for full models including all predictors and control variables.

different from zero for all measures of mindset and all indicators of health, providing support for the hypothesized causal pathway from mindset to health through approach coping and perceived stress. Therefore, we conclude that the data are consistent with the hypothesized causal pathway, that stress-is-enhancing mindsets lead to more approach coping in the face of stress, which in turn leads to lower levels of perceived distress, which in turn leads to better mental and physical health.

## Discussion

The overarching goal of this research was to evaluate how stress mindsets relate to health and performance in an undergraduate sample. First, we aimed to expand the current understanding of stress mindsets by exploring whether mindsets vary across specific sources of stress. To our knowledge, this is the first study to examine stress mindsets as a function of the perceived controllability and duration of sources of stress. Our results showed that mindsets about stress vary as a function of their source, and that participants viewed stress arising from controllable stressors as more enhancing than stress arising from uncontrollable stressors. Beliefs about stress did not vary as a function of whether stress arose from acute or chronic stressors.

Our second aim was to replicate prior work [11] suggesting that student mindset is related to academic performance and well-being. Consistent with prior findings, believing stress to be enhancing in general was related to better mental and physical health. This relationship was true for some, but not all, source-specific stress mindsets. In contrast to prior findings, stress mindsets were not associated with academic performance, as operationalized by self-reported GPA Third, we evaluated whether stress mindsets moderate the effects of stress on health, and found evidence that it does for some measures of health, but not others Finally, we explored approach coping and perceived distress as potential mediators of the relationship between stress and health. We found that believing stress to be enhancing was related to greater use of approach coping and lower perceived distress, which in turn was related to better health.

In the sections that follow, we review the broader implications of these findings for understanding stress mindsets and their consequences. Along the way, we discuss the limitations of this work and offer future directions.

## Understanding stress mindsets and their consequences

**Stress mindsets vary based on the controllability and duration of the source.** Our findings broaden the understanding of stress mindset theory as outlined in previous research [4, 6, 42]. Prior research has compared mindsets about stress in general to mindsets about a significant current stressor, finding that the two stress mindsets were closely related [4]. Our study is the first to explore whether mindsets are stable across other specific sources of stress, examining varying categories of stressor-type. The finding of weak correlations between mindsets for general and specific sources of stress suggests that general stress mindsets do not necessarily predict valuations of stress as it arises from varying sources.

Consistent with our predictions, students viewed stress as more enhancing when it arose from controllable than uncontrollable sources. This finding is consistent with Lazarus and Folkman's [28] framework of the stress response—sources of stress that people feel capable of handling (i.e., stressors that are viewed as controllable) are interpreted as "challenges" rather than "threats" and result in more productive stress responses (e.g., problem-solving rather than avoidance). However, in contrast to our prediction, students did not view stress arising from chronic stressors as more debilitating than from acute stressors. This lack of difference suggests that students' beliefs about stress may be shaped more by the extent to which they feel they can do something to respond to a stressor (e.g., studying) than by the duration of the stressor. It is possible, however, that duration of the source of stress did not affect stress mindset because of the specific situations students were asked to consider. Stress arising from the *acute* uncontrollable situation of waiting for a test grade was viewed by students as particularly debilitating. By contrast, stress arising from the *chronic* uncontrollable stressor of being in a class where the professor randomly calls on students may have been viewed as more enhancing, perhaps because, although the vignette specified that preparation would not help the situation, students may still expect that this source of stress is somewhat manageable with extra studying. Future research can build on these initial findings by presenting students with a more extensive set of vignettes, ideally rigorously pilot-tested (unlike those used in the present study), that present a variety of stressful situations varying in controllability and duration, and investigating more deeply the features of stressors that lead stress to be viewed as enhancing versus debilitating.

**General and source-specific stress mindsets predict health but not academic performance.** Overall, the chronic controllable mindset was most pervasively related to measures of health in our study. Chronic controllable stress was exemplified by the situation of having a quiz every class (chronic) for which one could prepare (controllable). General and controllable mindsets showed stronger correlations to measures of health, coping, and perceived distress than did uncontrollable mindsets. These findings suggest that adopting a stress-is-enhancing mindset may be more or less beneficial depending on the source of one's stress. When it comes to stress arising from controllable stressors (stress that something can be done about), a more enhancing mindset may encourage more productive coping, thus reducing perceived distress, and the impact of stress on physical and mental health.

Previous research has found a relationship between stress mindset and academic performance [11] such that believing general stress to be enhancing was related to a higher GPA. In the present study, we found associations in a similar direction, but direct effects were modest and non-significant. One challenge in identifying an association between stress mindset and GPA in the current study is the nature of our samples. One sample represented high-achieving students at a selective university who were relatively homogenous in GPA; the other sample came from a more diverse pool of online participants, who may come from institutions with considerable unaccounted variability in grading. A second challenge was that our measure of

GPA was retrospective, and thus an imperfect proxy of students' academic performance. Future work might consider more 'objective' measures of academic performance, such as institutional records data to assess GPA.

**Stress mindsets moderate the effect of stress on health.** This study examined stress mindsets as a moderator of the association between stress and health. Mindsets moderated the relationship between stress and health for all measures of health. Moderation depended on how stress was measured (perceived distress versus history of stressful life events) and was true for some mindsets, but not others. In general, consistent with hypotheses, stress-is-enhancing mindsets were related to better mental and physical health with increasing stress, as compared to stress-is-debilitating mindsets. Our findings suggest that stress has less of an impact on health for those who believe stress to be enhancing. In other words, if two people are experiencing the same level of perceived stress, those with a stress-is-debilitating mindset will experience a greater increase in symptoms of poor mental health, as compared to a person with a stress-is-enhancing mindset at the same level of stress. Similarly, if two people have experienced a similar amount of lifetime stress, the history of stressful events will take a greater toll on the person who believes stress to be more debilitating.

The one exception to this pattern was the effect of perceived distress on self-reported health as a function of general stress mindsets—in this case stress-is-enhancing mindsets were associated with *worse* perceived health with increasing stress. This finding—which warrants replication in future research given that it was in the opposite direction we predicted and emerged for only one measure of mindset—suggests that stress-is-enhancing mindsets may actually exacerbate the effects of stress on perceptions of health in some cases.

## Understanding how beliefs shape the consequences of stress

One of the guiding motivations of this research was to investigate how beliefs about stress influence health. Consistent with prior research, we found that stress-is-enhancing mindsets predicted better reported mental and physical health [4, 11]. We extended this previous work by evaluating a potential mechanism through which mindset influences health: We hypothesized that stress-is-enhancing mindsets would predict increased use of approach coping and decreased perceived distress, which, in turn, would promote better health.

Our findings were consistent with this hypothesis. In line with previous research [4, 11], we found that holding a stress-is-enhancing mindset was associated with higher endorsement of approach coping strategies and lower perceived distress. Approach coping and decreased perceived distress predicted better mental and physical health. These findings suggest that stress mindsets may influence health through changes in approach coping and perceived distress, with stress-is-enhancing mindsets promoting approach coping and decreasing levels of perceived distress, leading to better mental and physical health. It is important to emphasize, however, that the data in this study were collected at a single point in time, limiting our ability to infer causality. Furthermore mediation analyses hypothesizing longitudinal processes estimated with cross-sectional data can yield biased estimates [43], so it is crucial for future research to replicate and build on these promising findings by employing longitudinal and experimental research designs to provide stronger tests of the hypothesized causal associations.

## Specific stress mindsets outside of the academic domain

Because the present research focused on an undergraduate population, specific mindsets were assessed with regard to stress arising from academic sources. Whether our findings regarding specific stress mindsets extend to non-academic stressors, and to non-student populations,

remains to be seen. Future research should extend the concept of source-specific mindsets to other populations and other sources of stress to examine whether specific types of mindsets (e.g., mindsets for stress arising from controllable versus uncontrollable sources in social or health domains) seem to be particularly relevant for stress and health.

With our focus on academic sources of stress, we did not examine how beliefs about stress may vary depending on whether the source of stress is academic or interpersonal. This may be an important differentiation to make—students may be able to view stress resulting from academic pressure as enhancing and motivating, but not stress related to relationships with family, friends, romantic partners, and others. More importantly, it remains unclear whether perceiving interpersonal sources of stress as enhancing would have beneficial impacts on health and productivity. Because interpersonal sources of stress often arise from the actions of others, they may seem less controllable than academic sources of stress. Therefore mindsets about stress resulting from interpersonal demands may have attenuated links to health and productivity compared to mindsets about stress resulting from academic stressors. On the other hand, stress resulting from interpersonal sources of stress could be viewed as 'enhancing' in the sense that this stress could motivate engagement in relationship maintenance and repair strategies, which could lead to better mental and physical health indirectly through facilitating the development and maintenance of high-quality, satisfying interpersonal relationships (for a review see [44]).

## Implications for intervention: When and how is a stress-is-enhancing mindset truly helpful?

Understanding that some mindsets are more predictive of health and academic performance than others is an important step in developing population-relevant stress-mindset interventions. Beliefs about stress arising from chronic controllable sources were most pervasively related to health in our sample. These findings suggest that student-oriented stress mindset interventions should focus on controllable sources of stress, such as studying for exams and writing final papers, and emphasize how students can view their stress as motivation to engage in proactive behavior (e.g., starting the paper earlier). Indeed, prior interventions that have focused on helping students interpret stress arising from a specific controllable stressor as adaptive have been successful at improving students' emotional experience and performance [19, 24]. Critically, these prior interventions focused on helping students reappraise the physiological arousal that arises as part of the stress response, rather than beliefs about stress more broadly, and focused on stress arising from a specific controllable stressor: test-taking. Thus, future work might offer broader interventions targeting stress, rather than physiological arousal specifically, and focus on a wider array of controllable, academic sources of stress. Messages framing stress as adaptive targeted toward these controllable stress mindsets may help students harness their stress productively and healthily. The recent work of Keech and colleagues [9] is a promising first step: this study utilized an imagery-based intervention to encourage students to think about the stress in their life, what positive consequences it might have, and how they might experience these consequences. This intervention demonstrated that college students' stress mindsets can be modified through interventions focused on stress arising from relevant, student-specific stressors and, furthermore, its findings suggest that such interventions can improve students' coping, academic performance, and perceived distress. The imagery-based intervention, however, only significantly benefited students with high baseline levels of perceived distress; more research is needed to examine the impact of stress-mindset interventions on the average college student.

This imagery-based intervention differed from original stress-mindset interventions designed by Crum and colleagues [4] in that it focused on how stress *can be* enhancing, rather than how stress *is* enhancing. This differentiation raises the question of whether there are boundaries to the benefits of a stress-is-enhancing mindset: Are situations when believing stress is enhancing is not helpful, or may even be harmful? In our study, we failed to observe a relationship between beliefs about stress arising from chronic uncontrollable stressors and health. It is possible that this lack of association reflects some boundary conditions for when a stress-is-enhancing mindset can be helpful. This interpretation supports limiting a stress-is-enhancing mindset, thinking about how stress *can be*, rather than *is*, beneficial.

A second interpretation is that stress-mindset theory could be modified to broaden the picture of an enhancing mindset, rather than limit it. Instead of thinking of stress as beneficial because it "facilitates learning and growth" or "enhances performance and productivity" (items from the Stress Mindset Scale, [4]), what if stress were thought of as a signal from the body that communicates various adaptive messages, depending on the situation? Just as people know to eat when they feel hungry, and sleep when they feel tired, might they also think to take a step back and reflect or engage in self-care when feeling stressed? Particularly when responding to uncontrollable stressors, viewing stress as a signal (of importance or reflecting personal values), rather than something to "harness," could perhaps be beneficial in motivating proactive behavior. Focusing on what stress is telling you, rather than the experience of stress itself, may be useful in reappraising stress as an adaptive response. Indeed, the most recent generation of stress mindset interventions do take this broader view of how stress can be enhancing, and future research should examine whether and how these interventions change general and specific stress mindsets and how individuals respond to different types of stress [18].

## Conclusion

We found that students' beliefs about stress in the academic domain vary as a function of stressor type and that beliefs about stress that arise from controllable stressors may be more predictive of health than beliefs about stress from uncontrollable stressors. Our work supports the idea that holding a stress-is-enhancing mindset is related to better health and presents a potential mechanism for this relationship: mediation through greater use of approach coping and decreased perceived distress.

Our findings suggest that interventions encouraging students to think about their stress as enhancing may be useful in improving health. Our findings suggest that these interventions may also buffer students from the negative impacts of stress on health, especially if the interventions are stressor specific, rather than targeted to stress in general. Developing source-specific interventions may increase the effectiveness of intervention by improving believability and targeting stressors that are more pervasively related to health. These interventions are an important next step for research, with the potential to help students more healthfully and productively face the stress integral to their college experience.

## Supporting information

**S1 Fig. Hypothesized serial mediation model linking stress mindset to health through approach coping and perceived stress.**
(TIF)

**S1 Table. Gender differences in stress mindsets, perceived stress, stressful life events, coping, and mental and physical health.** Results are from four multivariate analyses of variance

examining differences in study variables as a function of gender and sample (for group differences by sample see S2 Table). Positive effect sizes indicate that the men were higher on a particular variable, negative effect sizes indicate that the women sample was higher on a particular variable.
(PDF)

**S2 Table. Sample differences in mindsets, perceived stress, stressful life events, coping, and mental and physical health.** Results are from four multivariate analyses of variance examining differences in study variables as a function of gender and sample (for group differences by gender see S1 Table). Positive effect sizes indicate that the participant pool sample was higher on a particular variable, negative effect sizes indicate that the MTurk sample was higher on a particular variable.
(PDF)

**S3 Table. Indirect effects of general stress mindsets on health through approach coping and perceived stress.**
(PDF)

**S4 Table. Indirect effects of specific stress mindset (acute controllable) on health through approach coping and perceived stress.**
(PDF)

**S5 Table. Indirect effects of specific stress mindset (chronic controllable) on health through approach coping and perceived stress.**
(PDF)

**S6 Table. Indirect effects of specific stress mindset (acute uncontrollable) on health through approach coping and perceived stress.**
(PDF)

**S7 Table. Indirect effects of specific stress mindset (chronic uncontrollable) on health through approach coping and perceived stress.**
(PDF)

**S1 Appendix. Adherence to preregistered analysis plan.**
(DOCX)

**S2 Appendix. Supplemental preregistered analysis.** Supplemental preregistered analysis examining whether stress mindsets are associated with health, controlling for history of stressful life events.
(DOCX)

**S3 Appendix. Confirmatory factor analysis to examine factor structure of the adapted stress mindset measures.**
(DOCX)

## Acknowledgments

The authors gratefully acknowledge Dr. Frank J. Keefe and the students of the Duke BRITElab for their thoughtful input on this project in its earlier stages.

## Author Contributions

**Conceptualization:** Anna Jenkins, Molly S. Weeks, Bridgette Martin Hard.

**Data curation:** Anna Jenkins.

**Formal analysis:** Anna Jenkins, Molly S. Weeks.

**Funding acquisition:** Anna Jenkins, Bridgette Martin Hard.

**Investigation:** Anna Jenkins.

**Methodology:** Anna Jenkins, Molly S. Weeks, Bridgette Martin Hard.

**Project administration:** Bridgette Martin Hard.

**Supervision:** Molly S. Weeks, Bridgette Martin Hard.

**Visualization:** Anna Jenkins, Molly S. Weeks.

**Writing – original draft:** Anna Jenkins, Molly S. Weeks, Bridgette Martin Hard.

**Writing – review & editing:** Anna Jenkins, Molly S. Weeks, Bridgette Martin Hard.

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
