## [Decision Letter · Decision Letter 0]

10 Jun 2021

PONE-D-21-03108

General and specific stress mindsets: Links with college student health and academic performance

PLOS ONE

Dear Dr. Hard,

Thank you for submitting your manuscript to PLOS ONE. After careful consideration, we feel that it has merit but does not fully meet PLOS ONE’s publication criteria as it currently stands. Therefore, we invite you to submit a revised version of the manuscript that addresses the points raised during the review process.

We look forward to receiving your revised manuscript.

Kind regards,

Zhidan Wang, Ph.D

Academic Editor

PLOS ONE

Journal Requirements:

2. Peer review at PLOS ONE is not double-blinded (https://journals.plos.org/plosone/s/editorial-and-peer-review-process). For this reason, authors should include in the revised manuscript all the information removed for blind review, including names of IRBs.

Reviewers' comments:

Reviewer's Responses to Questions

**Comments to the Author**

1. Is the manuscript technically sound, and do the data support the conclusions?

Reviewer #1: Partly

Reviewer #2: Yes

2. Has the statistical analysis been performed appropriately and rigorously? 

Reviewer #1: No

Reviewer #2: Yes

3. Have the authors made all data underlying the findings in their manuscript fully available?

Reviewer #1: Yes

Reviewer #2: Yes

4. Is the manuscript presented in an intelligible fashion and written in standard English?

Reviewer #1: No

Reviewer #2: Yes

5. Review Comments to the Author

Reviewer #1: The aspect of this research that compares stress mindset across specific sources/stressor types begins to fill a key gap in knowledge regarding stress mindset. Unfortunately, this key result seems to get lost in all of the other tests and results being reported. Restructuring the paper to center around this finding may be beneficial. Further, this is the largest paper reporting a single cross-sectional study that I have ever seen, and in my view, there is no need for the paper to be this big. I recommend making the entire paper more concise and aiming for less than 30 pages all inclusive. The measurement of GPA and the lack of information about validity also appear to be major limitations of this research.

Abstract

•Please include reference to study design (i.e., cross-sectional correlational). The specific design type should also be mentioned in the main Method section.

Introduction

•A key strength of the paper it its adherence to open science practices, most notably, pre-registration. I recommend including a statement of whether the study and analyses were carried out in accordance with the registered protocol, or whether there were changes. I note that in the pre-registration there is mention that some data were already collected at the time of registration. More detail on this is needed.

Method

•Validity information (and not just reliability) is required for all measures.

•The 3-item version of the stress mindset measure general needs stronger justification. I cannot see how saving 5 items from a survey is enough of a benefit to drop more than half of the items of a validated scale. You would need to report validity information (e.g., CFA and other types of validity) to give confidence in this. Did the specific mindsets also use 3 items each? Which items were they?

•The lack of pilot test of the vignettes is a limitation that should be acknowledged.

•Self-report physical health: The HRQOL that you have used contains items about mental health, but you refer to this as a physical health measure. Usually, you would just use the physical health items if making such a claim.

•Coping: For two item scales, Spearman Brown (and not alpha) should be used. Further, while you cite that Crum et al. (2013) created four composites, however, this does not suggest that it is a valid or useful approach – just that another research once did it. Again, validity information would be required (especially CFAs).

•Predicting self-reports of past GPA does not seem valid. You are using stress today, and/or stress mindset today, to predict academic performance last semester.

Results

•The very low correlations between stress mindset and source specific stress mindsets are not consistent with prior literature (although I acknowledge that in prior literature they were not separated in this manner). However, you have made variations to scales and validity remains unclear. I wonder whether this is an actual difference or something attributable to psychometrics. This should be discussed.

•When discussing significant effects (e.g., MANOVAs on page 26), it would be good to be specific about the effect size interpretation. Any conclusion of differences based on significantly different specific mindsets should be qualified by the size of the effect to ensure that small effects are not over-interpreted.

•I suggest avoiding the use of “marginally significant”.

•The study reports an enormous number of tests. To guard against type 1 error, I recommend adopting a more conservative alpha criterion (e.g., 0.01 or .001).

•“stressful life events is not an explanatory variable in the link between stress mindsets and health.” – Here and in places in the results section this description is used and I find it confusing given the associations tested. This would suggest life events is a moderator or mediator of the relationship between stress mindset and health.

•“Interestingly, general stress mindset was not uniquely associated with any of the health outcomes once the source-specific mindsets were taken into account” – this actually seems very logical. Presumably a general mindset would comprise beliefs about different types of stressors. If different types are measured and accounted for, the general mindset shouldn’t add anything further.

•Page 40: Both mediation and buffer are terms that are used. Which is it?

Discussion

•Some of the titles in the Discussion over-simplify the findings in a way that is problematic given people often quickly glance through articles. For example, the title on page 47 is very straightforward, but the results are more nuanced.

•Page 49: Not only do cross-sectional designs mean that causality cannot be inferred, they also provide biased estimates of parameters in mediation.

Reviewer #2: This article extends a recent line of research that reports the distinct impact of beliefs about stress. While currently published studies have focused on participants’ general mindset about stress, this study additionally examines beliefs about acute vs. chronic and controllable vs. uncontrollable stress. Overall the study is clearly described, the analyses are thorough and described in detail, and the manuscript explores some implications of the findings. Below, I note a few issues for the authors to address, but overall the study was effective in addressing the guiding research questions.

The introduction is organized clearly and reviews the guiding questions/sub-questions of the study. However, the authors only provide hypotheses for some of the goals listed on page 12. What were the authors’ hypotheses for c and d?

Table 4, Step 2 indicates that stressful life events is a significant, unique predictor but the regression coefficient and its standard error are 0. This is difficult to understand without clarification. On a related note, the separate section reporting history of stressful life events as a potential explanatory variable (page 36 – 37) seems unnecessary because it is highly redundant with the hierarchical regression models presented on pages 28 -33.

The study, and particularly the mediation analyses, are limited by the cross-sectional design. This is a limitation to the study and its conclusions, but there is not much that can added beyond what the authors note about this issue.

6. PLOS authors have the option to publish the peer review history of their article (what does this mean?). If published, this will include your full peer review and any attached files.

Reviewer #1: No

Reviewer #2: No

---

## [Author Response · Author response to Decision Letter 0]

23 Jul 2021

July 23, 2021

Dear Dr. Wang,

Thank you for the opportunity to revise and resubmit our manuscript “General and specific stress mindsets: Links with college student health and academic performance” for consideration in PLOS ONE. The reviews were exceptionally helpful, and we appreciate the opportunity to revise our manuscript based on this feedback. We believe the manuscript is now much stronger and more clearly conveys our findings. 

• As requested, we have submitted a marked copy of the manuscript with all changes tracked, labeled 'Revised Manuscript with Track Changes'. We have also submitted an unmarked version, marked “Manuscript”. 

• We have double-checked that our manuscript conforms to PLOS ONE style requirements, including those for file naming. 

• We have added in any information that was previously removed assuming masked review, specifically the name of the Institution that approved the IRB. Please note that we do not use the University name in referencing the participant pool sample (we still say “mid-sized selective private university in the southeastern U.S.” as this is specified in our IRB-approved protocol).

• We have included captions for our Supporting Information files at the end of the manuscript and updated any in-text citations to match accordingly. 

We have responded to each of the reviewers’ points below, which we have grouped by reviewer. 

Sincerely,

The Authors

Responses to Reviewers

Reviewer 1:

REVIEWER COMMENT: The aspect of this research that compares stress mindset across specific sources/stressor types begins to fill a key gap in knowledge regarding stress mindset. Unfortunately, this key result seems to get lost in all of the other tests and results being reported. Restructuring the paper to center around this finding may be beneficial. Further, this is the largest paper reporting a single cross-sectional study that I have ever seen, and in my view, there is no need for the paper to be this big. I recommend making the entire paper more concise and aiming for less than 30 pages all inclusive. The measurement of GPA and the lack of information about validity also appear to be major limitations of this research.

RESPONSE: We were very glad to see that the reviewer appreciated the key gap in the literature that our study can fill, and we certainly acknowledge that there are a lot of results for the reader to wade through. We have tried to strike a balance between making our manuscript readable and also reporting all of the analyses that we pre-registered. Looking at the manuscript again, we thought that we could reasonably move one of our more tangential pre-registered analyses—the analysis examining whether history of stressful life events is an explanatory variable in the link between stress mindsets and health—to an Appendix, along with an associated table. Given the challenges associated with our GPA measure (addressed more thoroughly in a later comment), we also decided to remove the exploratory analysis examining approach coping and perceived distress as potential mediators in the relationship between stress mindset and GPA. This has reduced the manuscript length by about 6 pages. We have also worked to streamline and reduce redundancy throughout the paper to improve its readability. We have attempted a much shorter manuscript before but were criticized by reviewers for not reporting all pre-registered analyses. A key motivation for submitting to PLOS ONE was that it does not have length restrictions and would allow us to more fully report all of our pre-registered analyses. We hope that our revisions to shorten the manuscript will be sufficient. 

We address the reviewer’s concerns over the measurement of GPA and lack of information about validity in a later point. 

REVIEWER COMMENT: (In reference to the abstract) Please include reference to study design (i.e., cross-sectional correlational). The specific design type should also be mentioned in the main Method section.

RESPONSE: We appreciate the need to clarify the nature of our study design. The abstract now describes the study in this way: “The goal of this cross-sectional, correlational study was to evaluate...” The methods section has also been revised to include this same information (p. 12: line 355 “The study employed a cross-sectional, correlational design.”

REVIEWER COMMENT: (in reference to the Introduction) A key strength of the paper is its adherence to open science practices, most notably, pre-registration. I recommend including a statement of whether the study and analyses were carried out in accordance with the registered protocol, or whether there were changes. I note that in the pre-registration there is mention that some data were already collected at the time of registration. More detail on this is needed.

RESPONSE: We have now added a document to the Open Science Framework called “Adherence to Preregistered Analysis Plan” that provides more details about adherence to the preregistration. As indicated in the analysis plan, some data for the study had already been collected at the time that the study was pre-registered. Specifically, we had recruited 449 out of the final 498 participants. However, none of the data were analyzed or even organized for analysis at the time of pre-registration. This document also describes one preregistered analysis that we have now moved into its own Appendix (S2 Appendix) (as a response to a previous comment by reviewer 1). Finally, they document describes the Confirmatory Factor Analysis that we have now added (as S3 Appendix) in response to another of Reviewer 1’s concerns, but that was not preregistered. 

REVIEWER COMMENT (In reference to the Method) 

• Validity information (and not just reliability) is required for all measures.

• The 3-item version of the stress mindset measure general needs stronger justification. I cannot see how saving 5 items from a survey is enough of a benefit to drop more than half of the items of a validated scale. You would need to report validity information (e.g., CFA and other types of validity) to give confidence in this. Did the specific mindsets also use 3 items each? Which items were they?

RESPONSE: Thank you for this feedback. As we described in the Measures section (p. 13-14, lines 455-412), we reduced the number of items in the stress mindset measure because participants would be completing these not only for general stress mindsets, but also for each of the four source-specific mindsets. This reduced the total stress mindset items from 40 to 15, which we thought was necessary to keep the overall survey at a manageable length. On p. 14, lines 423-425, we described the three items that were used: “Participants responded to the three items (i.e., “Experiencing stress facilitates my learning and growth,” “Experiencing stress debilitates my performance and productivity” [reverse-scored], “The effects of stress are positive and should be utilized”) on a 5-point scale (0 = strongly disagree, 4 = strongly agree).”

Although we did already work to establish the internal reliability of our three-item measure (see page 14, lines 427-434), we do appreciate the reviewer’s concerns over validity. We have conducted a confirmatory factor analysis with the five stress mindset scales (General Stress Mindset and the four Source-Specific Stress Mindset measures), which provides additional support for our measurement approach. The CFA is described briefly in the Method section on p. 17, lines 324–331, and in more detail in S3 Appendix.

We are open to the reviewer’s comment that validity information needs to be provided for all measures but are uncertain what the reviewer has in mind. We have described all measures consistently with other studies in the social psychological literature and have cited many papers in the manuscript that are relevant to establishing measurement validity. If the reviewer strongly feels this is insufficient, we are very open to specific guidance on what additional other ways to establish validity information should be provided. 

REVIEWER COMMENT: The lack of pilot test of the vignettes is a limitation that should be acknowledged.

RESPONSE: We have added mention to the lack of pilot testing in the general discussion on p. 40, lines 1101-1105: “Future research can build on these initial findings by presenting students with a more extensive set of vignettes, ideally rigorously pilot-tested (unlike those used in the present study), that present a variety of stressful situations varying in controllability and duration, and investigating.”

REVIEWER COMMENT: Self-report physical health: The HRQOL that you have used contains items about mental health, but you refer to this as a physical health measure. Usually, you would just use the physical health items if making such a claim.

RESPONSE: We appreciate the reviewer’s detection of this oversight. We have corrected all headings and mention of findings related to the HRQOL to “self-reported health” or to “mental and physical health”

REVIEWER COMMENT: Coping: For two item scales, Spearman Brown (and not alpha) should be used. Further, while you cite that Crum et al. (2013) created four composites, however, this does not suggest that it is a valid or useful approach – just that another researcher once did it. Again, validity information would be required (especially CFAs).

RESPONSE: We thank the reviewer for these suggestions. We have changed the reliability information presented for the two-item Brief COPE subscales from Cronbach’s alpha to Spearman-Brown coefficients. Please note that the point estimates for the Spearman-Brown coefficients were identical to the Cronbach’s alpha coefficients which we had previously reported when rounded to two decimal places, so although the labels have changed the point estimates are the same. This similarity in point estimates indicates that Cronbach’s alpha estimates weren’t significantly biased in this case, but we take the general point that the Spearman-Brown coefficient is generally a more appropriate reliability metric for two-item scales (Eisinga, Grotenhuis, & Pelzer, 2013). 

With regard to the four coping composites created based on Crum, Salovey, and Achor’s (2013) previous work, we think our approach here was appropriate given that we are attempting to replicate and build upon Crum et al.’s work, and also that examining the psychometric properties of the Brief COPE (Carver, 1997) was not a focus of our study. Crum et al. conducted a principal components analysis to identify the four higher-order coping composites in their sample and showed that the four composites were associated with various indicators of stress appraisal (including perceived distress), and with stress mindsets (see Crum et al., 2013, pp. 719–721). In our revision (lines 581-582), we now refer the reader to this specific location in the Crum et al. paper for more validity information. This is not to say that additional investigation of the psychometric properties and validity of the Brief COPE is not warranted, just that we view such an investigation as beyond the scope of the present manuscript.

REVIEWER COMMENT: Predicting self-reports of past GPA does not seem valid. You are using stress today, and/or stress mindset today, to predict academic performance last semester.

RESPONSE: We greatly appreciate this concern. Cumulative GPA can only be calculated based on the last semester in which a student was enrolled and received grades, as students don’t yet know their grades for the current semester in which they are enrolled. This makes the measure imperfect, but it is still a reasonable proxy for students’ current level of academic performance. We do acknowledge several limitations with the GPA measure and have added this specific concern to the General Discussion (p. 41, line 1130-1132: “A second challenge was that our measure of GPA was retrospective, and thus an imperfect proxy of students’ academic performance. Future work might consider more ‘objective’ measures of academic performance, such as institutional records data to assess GPA.” We have removed the mediation analysis associated with GPA and have been sure to report correlations as non-significant, rather than “marginally significant,” but would prefer to include it given that it was part of our pre-registration and relevant given prior findings by Keech et al. (2018).

REVIEWER COMMENT: (Regarding the results) The very low correlations between stress mindset and source specific stress mindsets are not consistent with prior literature (although I acknowledge that in prior literature they were not separated in this manner). However, you have made variations to scales and validity remains unclear. I wonder whether this is an actual difference or something attributable to psychometrics. This should be discussed.

RESPONSE: Thank you for raising this point. We hope that the addition of the CFA further documenting the psychometric properties of the 3-item general and source-specific stress mindset measures gives the reader further information about the validity of the scales. Although the correlations among the latent general and source-specific stress mindset factors presented in Supplement S3, Figure A are generally stronger than among the observed means (given the removal of unshared sources of error from the latent factors), they are still in the same ballpark, and paint a general picture of modest to negligible associations between general and source-specific stress mindsets.

REVIEWER COMMENT: When discussing significant effects (e.g., MANOVAs on page 26), it would be good to be specific about the effect size interpretation. Any conclusion of differences based on significantly different specific mindsets should be qualified by the size of the effect to ensure that small effects are not over-interpreted.

RESPONSE: Thank you for this comment. We had previously included some commentary on effect size and have tried to do so more consistently throughout the manuscript in this revision. Also, we are careful to report effect sizes in the form of correlation coefficients, overall variance explained, and squared semi-partial correlation coefficients, in addition to partial η2 when reporting MANOVAS.

REVIEWER COMMENT: I suggest avoiding the use of “marginally significant”.

RESPONSE: We have removed all reference to any results as “marginally significant.”

REVIEWER COMMENT: The study reports an enormous number of tests. To guard against type 1 error, I recommend adopting a more conservative alpha criterion (e.g., 0.01 or .001).

RESPONSE: We appreciate this point. Currently, we report whether all p-values are < .05, < .01, or < .001. The vast majority of significant results have p < .01 or .001. Rather than adjust all of our analyses, our preference is to leave this information in, along with effect size information, and allow readers to evaluate it for themselves. 

REVIEWER COMMENT: “stressful life events is not an explanatory variable in the link between stress mindsets and health.” – Here and in places in the results section this description is used and I find it confusing given the associations tested. This would suggest life events is a moderator or mediator of the relationship between stress mindset and health.

RESPONSE: As noted previously, we have now moved this analysis into the appendix to conserve space. By “explanatory variable,” we meant a third variable, not a moderator or mediator. 

REVIEWER COMMENT: “Interestingly, general stress mindset was not uniquely associated with any of the health outcomes once the source-specific mindsets were taken into account” – this actually seems very logical. Presumably a general mindset would comprise beliefs about different types of stressors. If different types are measured and accounted for, the general mindset shouldn’t add anything further.

RESPONSE: This does seem logical, however the small correlations between the source-specific mindsets and the general mindset seems to suggest against this interpretation.

REVIEWER COMMENT: Page 40: Both mediation and buffer are terms that are used. Which is it?

RESPONSE: Thank you for this point of clarification. As described in the Introduction (pp. 10–11) and on page 35, based on previous literature we hypothesize that both processes may be at play. We state: As we described on this page (now p. 35) we think it may be BOTH. “In addition to buffering against the negative effects of perceived distress on health, stress mindset might also directly improve health by reducing perceived distress, specifically by increasing the use of approach coping strategies.” In other words, we think that stress mindsets may affect how people respond to perceived distress, thereby moderating its effects on health, but that it may also directly affect and reduce the experience of perceived distress by changing coping strategies, a mediating effect.

REVIEWER COMMENT: (regarding the Discussion): Some of the titles in the Discussion over-simplify the findings in a way that is problematic given people often quickly glance through articles. For example, the title on page 47 is very straightforward, but the results are more nuanced.

RESPONSE: We appreciate this comment and have worked to revise the headers to be a bit more detailed and yet appropriately concise. The new current headers will certainly still oversimplify some nuance, but we believe they aren’t misleading.

REVIEWER COMMENT: Page 49: Not only do cross-sectional designs mean that causality cannot be inferred, they also provide biased estimates of parameters in mediation.

RESPONSE: Thank you for raising this issue. We have now added a reference to Maxwell, Cole, and Mitchell (2011) documenting biased parameter estimates in cross-sectional designs and continue to frame our exploratory mediation analyses as simply a first step in examining this potential process model of the link between stress mindsets and health.

Reviewer 2:

REVIEWER COMMENT: This article extends a recent line of research that reports the distinct impact of beliefs about stress. While currently published studies have focused on participants’ general mindset about stress, this study additionally examines beliefs about acute vs. chronic and controllable vs. uncontrollable stress. Overall the study is clearly described, the analyses are thorough and described in detail, and the manuscript explores some implications of the findings. Below, I note a few issues for the authors to address, but overall the study was effective in addressing the guiding research questions.

The introduction is organized clearly and reviews the guiding questions/sub-questions of the study. However, the authors only provide hypotheses for some of the goals listed on page 12. What were the authors’ hypotheses for c and d?

RESPONSE: Although we state our hypothesis for goal (c) (evaluate whether or not mindsets moderate the effects of stress on health) in the results section, we appreciate the reviewer pointing out that this was not clear from the introduction. Our hypothesis was that viewing stress as enhancing would buffer the negative relationship between stress and health. We have now added a few sentences to the introduction (page 9 and 10, lines 195-205) to preview this prediction, and then have made it clearer in specifying our goals. 

As noted earlier, we have moved question (d) out of the manuscript to the Appendix. 

REVIEWER COMMENT: Table 4, Step 2 indicates that stressful life events is a significant, unique predictor but the regression coefficient and its standard error are 0. This is difficult to understand without clarification. On a related note, the separate section reporting history of stressful life events as a potential explanatory variable (page 36 – 37) seems unnecessary because it is highly redundant with the hierarchical regression models presented on pages 28 -33.

RESPONSE: Thank you for raising this important issue. We have revised the entries in Table 4 related to history of stressful life events so the parameter estimates are reported out to more decimal places. The very small values for the regression coefficients for history of stressful life events have to do with the scaling of the weighted stressful life events measure, which is a sum score of previously experienced stressful life events, each weighted on a scale from 0–100. Weighted sum scores range from 0 to 1516 in this sample. The interpretation of the regression coefficient is the unit increase in the outcome associated with each unit increase in stressful life events, and the units for the weighted stressful life events scale are very small.

As noted previously, we have moved information about history of stressful life events as a potential explanatory variable to the Appendix. 

REVIEWER COMMENT: The study, and particularly the mediation analyses, are limited by the cross-sectional design. This is a limitation to the study and its conclusions, but there is not much that can added beyond what the authors note about this issue.

RESPONSE: We understand and certainly agree with this concern. We hope we have done enough to be forthright about this limitation.

---

## [Editor Report · Decision Letter 1]

5 Aug 2021

General and specific stress mindsets: Links with college student health and academic performance

PONE-D-21-03108R1

Dear Dr. Hard,

We’re pleased to inform you that your manuscript has been judged scientifically suitable for publication and will be formally accepted for publication once it meets all outstanding technical requirements.

Kind regards,

Zhidan Wang, Ph.D

Academic Editor

PLOS ONE
---

## [Editor Report · Acceptance letter]

12 Aug 2021

PONE-D-21-03108R1 

General and specific stress mindsets: Links with college student health and academic performance 

Dear Dr. Hard:

I'm pleased to inform you that your manuscript has been deemed suitable for publication in PLOS ONE. Congratulations! Your manuscript is now with our production department. 

Kind regards, 

on behalf of

Dr. Zhidan Wang 

Academic Editor

PLOS ONE